# The Childbirth Fear Questionnaire and the Wijma Delivery Expectancy Questionnaire as Screening Tools for Specific Phobia, Fear of Childbirth

**DOI:** 10.3390/ijerph19084647

**Published:** 2022-04-12

**Authors:** Nichole Fairbrother, Arianne Albert, Fanie Collardeau, Cora Keeney

**Affiliations:** 1Department of Family Practice, University of British Columbia, Vancouver, BC V6T 1Z4, Canada; cora.keeney@ubc.ca; 2Women’s Health Research Institute, Vancouver, BC V6H 2N9, Canada; arianne.albert@cw.bc.ca; 3Department of Psychology, University of Victoria, Victoria, BC V8P 5C2, Canada; faniecol@uvic.ca

**Keywords:** perinatal mental health, anxiety disorders, perinatal anxiety, fear of childbirth, screening

## Abstract

Background: Perinatal anxiety and related disorders are common (20%), distressing and impairing. Fear of childbirth (FoB) is a common type of perinatal anxiety associated with negative mental health, obstetrical, childbirth and child outcomes. Screening can facilitate treatment access for those most in need. Objectives: The purpose of this research was to evaluate the accuracy of the Childbirth Fear Questionnaire (CFQ) and the Wijma Delivery Expectations Questionnaire (W-DEQ) of FoB as screening tools for a specific phobia, FoB. Methods: A total of 659 English-speaking pregnant women living in Canada and over the age of 18 were recruited for the study. Participants completed an online survey of demographic, current pregnancy and reproductive history information, as well as the CFQ and the W-DEQ, and a telephone interview to assess specific phobia FoB. Results: Symptoms meeting full and subclinical diagnostic criteria for a specific phobia, FoB, were reported by 3.3% and 7.1% of participants, respectively. The W-DEQ met or exceeded the criteria for a “good enough” screening tool across several analyses, whereas the CFQ only met these criteria in one analysis and came close in three others. Conclusions: The W-DEQ demonstrated high performance as a screening tool for a specific phobia, FoB, with accuracy superior to that of the CFQ. Additional research to ensure the stability of these findings is needed.

## 1. Introduction

Anxiety and anxiety-related conditions are the most prevalent of all psychiatric disorders [1,2]. A third of the adult population will suffer from one or more anxiety or anxiety-related disorder at some time in their life [1]. This is significantly greater than the prevalence of mood disorders (i.e., depressive and bipolar disorders) at 21.4% [1]. Women are also 1.5 times as likely as men to suffer from anxiety or anxiety-related condition [1,2]. A recent meta-analysis indicates that one in five pregnant and postpartum people suffer from one or more anxiety or anxiety-related disorder during pregnancy or postpartum [3]. This is significantly more than perinatal depression, where six to twelve percent of pregnant and postpartum people suffer from an episode of major depression during the perinatal period [4,5].

Anxiety and anxiety-related disorders are associated with substantial indirect costs related to functional impairment (e.g., diminished work capacity, unemployment) [4]. People with these conditions are significantly more impaired with respect to social, emotional and physical functioning compared with non-anxious individuals [6]. Anxiety and its related disorders are associated with high levels of health care service utilization [7,8,9,10,11].

Some level of maternal prenatal anxiety (i.e., dimensional anxiety not necessarily associated with a diagnosis) is a normal aspect of pregnancy for many, if not most, pregnant people and unlikely to negatively impact fetal or obstetric outcomes [12]. Maternal prenatal anxiety has, despite various methodological challenges and limitations [12,13,14,15], been associated with a number of adverse pregnancy outcomes such as preterm delivery, miscarriage, preeclampsia and low birth weight [12,16,17,18,19], as well as some negative effects on the developing infant, including small differences in brain development and attention, and small effects on infant temperament and emotion-regulation [12,13,14,15,20,21,22]. Prenatal maternal anxiety is also a strong risk factor for postpartum depression, even after controlling for prenatal depression [23,24,25,26]. Anxiety and their related disorders, specifically, were also found to be associated with deleterious fetal, infant and maternal outcomes, including pregnancy complications and preterm birth, spontaneous abortions, neonatal morbidity and lower birth weight [27,28,29,30,31]. For example, mothers with postpartum obsessive-compulsive disorder were found to be less confident and sensitive in mother-infant interactions than mothers without obsessive-compulsive disorder [32]. Additionally, maternal postpartum social anxiety disorder was associated with reduced cognitive and language abilities in offspring [33]. Overall, maternal anxiety disorders are predictive of anxiety disorders in offspring [34].

There are a number of domains of anxiety (i.e., content areas) that are a particular focus among perinatal people. These include obsessive compulsive disorder (OCD), in which the focus of the obsessions (a core feature of OCD) is on harm coming to one’s infant [35], post-traumatic stress disorder (PTSD) subsequent to traumatic childbirth [36], a fear of needles or other medical procedures (e.g., instrumental or surgical birth) [37], pregnancy-specific anxiety (i.e., high anxiety related to the wellbeing of one’s pregnancy [38]) and fear of childbirth (FoB) [39]. FoB is the focus of the current study.

FoB is common among people with childbearing potential (i.e., people who are pregnant, may become pregnant or who have already given birth). In the most comprehensive systematic review and meta-analyses of FoB in pregnant women conducted to date, the worldwide pooled prevalence of FoB was estimated at 14% (95% CI 0.12–0.16) [40]. The study was based on data from 29 primary studies and included a total of 853,988 pregnant women. Prevalence estimates from individual studies varied significantly from 3.7 to 43%. Of concern is that there was a high level of between-study heterogeneity, not explained via sensitivity and subgroup analyses. Unexplained variability in prevalence estimates may be a result of the significant methodological variability across studies (e.g., variability in cut-scores and measurement tools). Historically, FoB was not conceptualized as a diagnosable mental health condition but rather a form of dimensional psychological distress characterized by fear and anxiety and assessed via a self-report inventory [40]. When mental health difficulties are assessed using self-report questionnaires, prevalence estimates tend to be much higher than when formal diagnostic criteria are employed [41,42,43]. For example, all of the studies included in this meta-analysis of prevalence employed self-report questionnaires and not diagnostic interviews. The one study in which diagnostic criteria were clearly employed also, as expected, reported a much lower prevalence of FoB (3.7%) compared with the meta-analysis as a whole [39].

FoB can be highly distressing and associated with various psychosocial, mental health, obstetrical, childbirth and child-related outcomes [44,45,46,47]. For some, FoB is so intense as to lead to delaying or avoiding pregnancy and pregnancy termination, even among those who wish to bear children [48,49,50]. Obstetrical and birth complications include increased requests for epidural anesthesia during labor [49,50], longer labors [51,52,53] and a higher likelihood of emergency and planned cesarean section (CS) [52,54,55,56,57,58,59]. For example, fear of vaginal birth is consistently associated with a preference for cesarean birth, and severe fear of vaginal birth has been associated with a greater likelihood of a cesarean birth without medical indications [52,60,61,62].

There is also a higher likelihood of negative birth experiences among women with a fear of childbirth [63,64], especially if the woman delivers by emergency CS or instrumental vaginal delivery [51,65]. There is also an association between FoB and mental health difficulties, including postnatal depression, specific phobia and PTSD [66,67,68,69]. In particular, there is a strong association between previous negative birth experiences and/or traumatic births and FoB [69]. History of prior operative or instrumental delivery was also associated with higher levels of FoB [57,58,70], with the odds of FoB increasing with the number of obstetric complications experienced during a previous pregnancy [65]. Women with a previous negative birth experience are five times more likely to experience FoB in a subsequent pregnancy [65]. Although most studies have found a positive relationship between parity and FoB, with higher levels of childbirth fear reported by nulliparous compared with multiparous women [45,47,50,58,70,71,72,73], there is some evidence that the most severe levels of FoB are experienced by multiparous women [39]. A range of socio-demographic variables are associated with higher levels of childbirth fear including lower educational attainment, younger age [74,75], low social support [61,76], dissatisfaction with partner or support received from partner [60,74], mental health variables such as higher anxiety and stress [54,57,60,72,74,76], history of depression and depression during pregnancy [61,74,77,78], low confidence in one’s ability to cope with labour and birth [61,74,77,78] and history of abuse [45,76,79]. Higher levels of fatigue during pregnancy [80] and lower self-rated health [81] were also associated with higher levels of FoB.

The lack of a clear diagnostic classification for FoB is problematic because, in the absence of diagnostic criteria, it may be difficult to determine which questionnaire-based cut-scores may best represent clinically meaningful fear meriting treatment. Specifically, to merit the diagnosis of an anxiety disorder, symptoms must be clinically distressing or functionally impairing [82]. Although not yet fully established, a specific phobia may be the most appropriate diagnostic category for FoB, in particular for nulliparous people [54,83,84,85]. A specific phobia is a fear and avoidance of circumscribed objects and situations (e.g., insects, animals, heights, blood, injections). Given that FoB is a circumscribed fear with symptoms and features closely resembling those of other specific phobias, it was proposed as perhaps the most appropriate diagnostic classification for FoB [54,83,84,85]. Further, tokophobia (severe FoB) is classified in the *International Classification of Diseases-11* as a phobic anxiety disorder [86]. Although other candidate disorders include PTSD (among multiparous people), health anxiety disorder, social anxiety disorder and generalized anxiety disorder, at present, the extant evidence suggests that specific phobia is a very reasonable place to start. In the only study to evaluate this systematically (N = 106), 8.5% of study participants (a general sample of nulliparous pregnant women in Sweden) were found to meet the Diagnostic and Statistical Manual of Mental Disorders, fifth edition (DSM-5) diagnostic criteria for specific phobia FoB [83]. Although small (N = 106), this is also the only study published to date to assess any self-report measure of FoB as a potential screening tool for diagnosable FoB [83]. In this study, a Wijma Delivery Expectancy Questionnaire (W-DEQ) score of ≥85 was found to be the optimal cut-off score for identifying FoB, with excellent sensitivity (100%), specificity (93.8%) and agreement between the W-DEQ A and the SCID-5 (specific phobia; Cohen’s Kappa coefficient, κ = 0.720). Determining appropriate cut-scores for self-report measures of FoB can be aided via studies in which diagnostic interviews for a specific phobia, FoB, were also employed, and screening metrics evaluated. In the absence of this, it is difficult to determine if cut-scores based on other approaches (e.g., the top 25% of scores) actually represent clinically meaningful distress and/or impairment in functioning. Given the above, we opted, in this study, to focus our attention on FoB diagnosable as a form of specific phobia.

Our study team recently developed a new measure of FoB: The Childbirth Fear Questionnaire (CFQ) [87]. The CFQ was designed to overcome the limitations of existing measures and as a screening tool for FoB. Existing measure frequently omit important domains of FoB [56,70,75,88,89,90,91,92,93,94,95], include non-fear related items [88,90,92,94,95,96,97], are too brief to encompass the full FoB experience (e.g., 1–2 items only) [56,70,75,91], or include too few items per subscale to achieve stability [92,98]. We developed the CFQ to cover the full range of domains of FoB with a view to enabling the identification of specific fear domains to be targeted in treatment. We also sought to develop a measure that would function well as a screening tool for diagnosable FoB. Screening represents a critical step in the pathway to treatment [99]. Although diagnostic assessments by trained professionals are the gold standard for providing mental health diagnoses, they are both expensive and time-consuming. Consequently, more rapid and cost-effective screening is essential for identifying those suffering from clinically meaningful FoB. Without screening, those suffering may fail to be identified and, as a result, fail to receive evidence-based care [100]. The CFQ was evaluated in two separate samples, with both an exploratory and a confirmatory factor analysis. The psychometric properties of the CFQ are strong, and two manuscripts pertaining to this measure were published, with a third currently under review [71,87,101].

The primary objective of this research was to evaluate the screening accuracy of the CFQ for subclinical and full criteria specific phobia, FoB. A secondary objective was to compare the screening accuracy of the CFQ to the screening accuracy of the W-DEQ. Given known differences in FoB between nulliparous and multiparous people [39,71,72], we also elected to report the screening accuracy of the CFQ and the W-DEQ separately for nulliparous and multiparous participants. As a further distinction, we also reported the accuracy of the CFQ and the W-DEQ separately for those primarily fearful of vaginal birth and those primarily fearful of cesarean birth. We hypothesized measures of FoB might perform differently for people whose primary fears relate to vaginal delivery compared to those whose primary fears relate to medical and surgical interventions (i.e., cesarean birth) [101]. We chose the W-DEQ as the comparator measure because: (a) the W-DEQ is the most commonly used measure to assess FoB and has broad international acceptance [46]; (b) the W-DEQ is the only measure of FoB to be evaluated as a screening tool for a specific phobia, FoB [46]; and (c) the CFQ was developed with a view of overcoming some of the limitations of the W-DEQ (i.e., the inclusion of non-fear-related items, and a failure to assess all of the relevant FoB content domains) [87]. In contrast with the W-DEQ, the CFQ assesses a broader range of FoB content areas, includes only fear-related items, and includes a measure of interference, making it more similar to a diagnostic measure (i.e., mental health diagnoses require either distress or interference in order for a diagnosis to be given).

## 2. Materials and Methods

This paper reports on a secondary analysis of a larger dataset, for which detailed methods were published [87].

### 2.1. Ethics

This research received ethical approval from the Behavioral Research Ethics Board of the University of British Columbia. All participants provided informed, written consent prior to participation.

### 2.2. Participants

All English-speaking, pregnant individuals over the age of 18 and residing in Canada were eligible to take part in this study. In total, 881 participants took part in the online questionnaire between 11- and 46-weeks’ gestation (an average of 35 weeks). Primary data collection took place between August 2016 and November 2019.

### 2.3. Procedures

Perinatal people were directed to the online survey via the study advertisement posted on online forums and social media pages frequented by pregnant women (e.g., pregnancy-related Facebook groups and websites). Participants who completed the survey were entered into a draw with the chance to win one of seven CAD 150 prizes.

### 2.4. Measures

Demographic (e.g., age, education, marital status, income, race/ethnicity and country of residence), pregnancy (e.g., number of fetuses and method of conception) and reproductive history (e.g., the number of prior pregnancies, births, miscarriages and vaginal and cesarian deliveries) was collected via self-report. Participants were also asked about their delivery preferences using a single question. Scoring for this item was based on a 7-point Likert-type scale ranging from “I have a very strong desire for a vaginal birth” (0) to “I have a very strong desire for a cesarian birth” (6). The center of the scale (3) was “I have no preference either way”.

The Childbirth Fear Questionnaire (CFQ) [87] is a recently developed, 40-item, self-report measure used to assess fear of childbirth. The 40 items are scored on a Likert-type scale ranging from 0 (no fear) to 4 (extreme fear), and measuring nine, frequently reported dimensions of FoB. The 40-item CFQ fear dimensions include (1) fear of loss of sexual pleasure or attractiveness (SEX), (2) fear of pain from a vaginal birth (PAIN), (3) fear of medical intervention (INT), (4) fear of embarrassment (SHY), (5) fear of harm to the baby (HARM), (6) fear of cesarean birth (CS), (7) fear of mom or baby dying (DEATH), (8) fear of insufficient pain medication (MEDS), (9) fear of body damage from a vaginal birth (DAMAGE). The dimensions are scored by taking the average of the item scores within that dimension (range = 0–4). The CFQ also includes an additional 8-item Interference scale with items covering multiple life domains. For the Interference scale, participants are asked to rate, from 0 (no interference) to 4 (extreme interference), how much their FoB interfered with various aspects of their life. Each of the eight items asked about interference with a different life domain (i.e., interference with one’s relationships with one’s partner/spouse, family members, prenatal caregivers and others, as well as interference with one’s work life, leisure activities and preparation for the new baby). The CFQ total score includes only the 40 fear items and is scored as the mean of the subscale scores (range = 0–4). The Interference scale is scored separately. Consequently, the CFQ produces a fear score and an interference score. Initial validation of the CFQ produced a Cronbach’s alpha reliability coefficient of 0.94 for the overall scale and a range between 0.76 and 0.94 for the individual subscales [71]. The CFQ demonstrated good convergent and discriminant validity when comparing the associations between the CFQ with other measures of FoB. Evidence suggests that the CFQ is accurate in detecting group differences between pregnant people in relation to delivery mode preference and parity.

The Wijma Delivery Expectancy Questionnaire (W-DEQ-A) [90]. The W-DEQ-A is a 33-item questionnaire. Items are scored on a 0–5 Likert type scale ranging from 0 (extremely) to 5 (not at all). The minimum and maximum scores of the questionnaires are 0 and 165, with higher scores reflecting higher levels of fear. The psychometric properties of the W-DEQ-A are well established [94,102]. The internal consistency reliability in the present sample was 0.92. In addition to the W-DEQ-A total score, there are data to support the administration of a 6-item fear scale, which were found to be highly correlated with the full scale and several other important outcomes [103].

The Diagnostic Assessment Research Tool (DART v1.03.16) [104]. The DART (v1.03.16) is a modular, semi-structured interview designed for the assessment of DSM-5 diagnoses. Although the DART remains early in its development, psychometric evidence to date strongly supports the interrater reliability and construct (convergent and discriminant) validity of the measure as a diagnostic interview for DSM-5 disorders [105]. We used the specific phobia section of the DART to assess specific phobia, fear of childbirth, in this study. Minor wording modifications were made to orient the interview exclusively to fear of childbirth. Interviewers were research assistants, graduate students in clinical psychology and the principal investigator, and were trained and supervised by the principal investigator. Participants’ responses were classified as indicating full criteria diagnosis, a subclinical diagnosis, or no diagnosis of specific phobia, FoB. Subclinical diagnoses are those in which all disorder criteria are endorsed other than the distress/impairment criteria (i.e., the symptoms do not cause clinically significant distress or life impairment). In the context of specific phobia, FoB, this implies that those who reported symptoms meeting the criteria for a subclinical specific phobia reported high levels of consistent and persistent fear of childbirth, but these symptoms failed to cause clinically significant distress or impairment in functioning. Because FoB appears to exist on a continuum from mild (i.e., most pregnant people experience some, at least low levels, of FoB) to severe (for some, it may be debilitating), it may be important to identify and offer services to pregnant people with subclinical levels of specific phobia, FoB, as well as those who report symptoms meeting full diagnostic criteria.

### 2.5. Data Analysis Strategy

All analyses were carried out in R v.4.1.1 [106] and SPSS v.24 [107].

The precision of estimates of a diagnostic accuracy study depends on the prevalence of the condition in the sample [108]. The lower the prevalence, the larger the number of participants with cases needed to precisely estimate metrics such as sensitivity and specificity, as lower prevalence results in estimated metrics that can be unreliable and imprecise [109]. For these reasons, we conducted an assessment of screening accuracy for both subclinical and full criteria diagnoses of specific phobia, FoB. Specifically, we began by comparing cases with a diagnosis meeting the full criteria for a specific phobia, FoB, to the remainder of the sample. However, due to small numbers of cases meeting full criteria, we also compared cases of full and subclinical criteria to the remainder of the sample.

Given the data indicating that childbirth fears may differ among nulliparous and multiparous people [72], we felt it was important to provide information about screening accuracy for each group separately. We have also provided screening accuracy data for the CFQ (total scores) with and without the Interference subscale included. We sought to investigate whether the interference subscale would improve screening accuracy. Screening accuracy was determined by using cutpoints of the scales to identify participants likely to have a specific phobia, FoB, compared to the results of the Diagnostic Assessment Research Tool for each participant.

To determine optimal cutpoints, we used the “cutpointr” [110] package in R. Cutpoints were estimated by maximizing the Youden’s J index using 1000 bootstrap replicates. The returned optimal cutpoint and its associated area under the curve (AUC), sensitivity, specificity, Youden’s J index, negative predictive value (NPV) and positive likelihood ratio (LR+) were the means of these metrics across all 1000 replicates. This whole process was bootstrapped 100 times to validate the out-of-sample performance. These “out of bag” or oob estimates are reported in the Results. To evaluate if specific combinations of items might be better predictors, we also used logistic regressions of each outcome (subclinical and full criteria diagnoses of specific phobia, FoB, and for those primarily fearful of vaginal birth and those primarily fearful of a cesarean birth) against all of the CFQ subscales. Non-significant subscales (*p* < 0.1) were removed from the models, and model predictions in the form of probabilities between 0 and 1 were calculated for each participant. These predicted probabilities were then subjected to the same cutpoint analysis as the subscales described above. Predicted probabilities of FoB can be calculated from the estimated log-odds (β) using the formula below.
$$P\left(\mathrm{FoB}\right)=\frac{1}{1+e^{-\left(\beta_{0}+\beta_{1}x_{1}+\ldots \right)}}$$

For all assessments of screening accuracy, we also sought to evaluate the screening accuracy of the CFQ and the W-DEQ against the criteria for a “good enough” screening tool proposed by Fairbrother and colleagues [111]. They propose that, in order for a screening tool to be deemed sufficiently accurate for use in clinical settings, it should meet certain minimum standards of accuracy, including an AUC of 0.8 or greater, a Youden’s J index of 0.5 or more (J = 0.05 when sensitivity and specificity both equal 0.75), a NPV of 0.8 or greater, and a LR+ of 4.0 or more. An LR+ of 4.0 means that with a positive test result, one is 25% more likely to have the condition in question compared with the baseline probability of having the condition [112]. Any recommendations regarding the accuracy and clinical utility of the CFQ and the W-DEQ is based on how well they perform in relation to these criteria.

## 3. Results

### 3.1. Participants

A total of 659 pregnant people participated in Subclinical and full criteria diagnoses of specific phobia in this study. Participants ranged in age from 21 to 49 (M = 32.9, SD = 4.10). Of these, 270 (48%) were nulliparous at the time of participation, and 296 (52%) were multiparous. Information pertaining to participant demographics, current pregnancy and reproductive history is provided in Table 1. Means and standard deviations for the CFQ and the W-DEQ are reported in Table 2.

### 3.2. Prevalence of Specific Phobia, Fear of Childbirth

Twenty-two (3.3%) participants reported symptoms meeting full diagnostic criteria for a specific phobia, fear of childbirth, and 47 (7.1%) reported symptoms meeting subclinical criteria for a specific phobia, fear of childbirth. When segregated by parity, fewer (1.9%) nulliparous participants met the full criteria for specific phobia compared with multiparous participants (5.1%). However, similar proportions of nulliparous and multiparous participants met subclinical criteria for a specific phobia, fear of childbirth (6.3 and 6.8%, respectively).

### 3.3. ROC Curves and Diagnostic Accuracy

We present the initial screening metrics for the CFQ and the W-DEQ in Table 3, Table 4 and Table 5, with corresponding ROC curves presented in Figure 1, Figure 2 and Figure 3 in the manuscript with supplementary ROC curves presented in the Appendix A.

In Table 3, screening metrics are provided for the CFQ (both with and without the Interference Subscale) and the W-DEQ for a specific phobia, FoB, full criteria across parity groups. In Table 4, we present the same findings, but for a specific phobia, FoB, full criteria and subclinical were combined. In Table 5, we present the screening metrics for the CFQ (including the Interference Subscale) and the W-DEQ across parity groups, separately for those primarily fearful of vaginal birth and those primarily fearful of cesarean birth. For this table, there were not enough cases to present the screening accuracy of the W-DEQ for fear of cesarean birth. Consequently, only the W-DEQ screening accuracy for fear of vaginal birth was provided. Given the smaller samples available for this final analysis, screening metrics are provided for subclinical and full diagnostic criteria cases combined.

In these preliminary ROC analyses, the W-DEQ evidenced the highest level of screening accuracy, meeting or exceeding the criteria for a “good enough” screening tool across several analyses. Specifically, when comparing those reporting symptoms meeting full diagnostic criteria for a specific phobia, FoB compared to the remainder of the sample, the W-DEQ met or exceeded the “good enough” criteria for both nulliparous and multiparous participants and came close to meeting these criteria for the full sample. When comparing those who reported symptoms meeting full or subclinical diagnoses with the remainder of the sample, the W-DEQ exceeded the criteria for a “good enough” screening tool for multiparous participants (in general and among those primarily fearful of a vaginal birth), as well as for all participants primarily fearful for a vaginal birth.

The CFQ only met or exceeded the criteria for a “good enough” screening tool for nulliparous participants primarily fearful of vaginal birth. When comparing those reporting symptoms meeting full or subclinical diagnoses with the remainder of the sample, the CFQ came close to meeting the criteria for a “good enough” screening tool for nulliparous participants in general, for nulliparous participants primarily fearful of cesarean birth, and for those primarily fearful of a vaginal birth (full sample).

However, cutpoints from the predicted probabilities of the logistic regressions performed much better for the sample as a whole and across nulliparous and multiparous participants separately. Specifically, among nulliparous participants, the INT, CS and Interference subscale emerged as significant predictors, resulting in screening metrics that exceeded the criteria for a “good enough screening tool”. The logistic regression predicting fear of vaginal birth (nulliparous participants only) included too few positive cases (*n* = 4) to accurately estimate logistic regression parameters. Fear of cesarean birth was predicted by the INT and CS subscales, with screening metrics again exceeding those required for a “good enough” measure. Among multiparous participants, diagnostic status was predicted by the CFQ SEX, PAIN and the Interference subscales. In this analysis, findings fell very slightly below those of a “good enough” measure (i.e., AUC = 0.84; Youden’s J Index = 0.42). Among multiparous participants with predominantly a fear of vaginal birth, SEX, PAIN, HARM, CS, DEATH and the Interference scale significantly predicted diagnostic status. In this case, the screening metrics exceeded the requirements of a “good enough” screening tool. In the case of participants primarily fearful of cesarean birth, only SEX and the Interference subscale significantly predicted diagnostic status. Screening metrics fell slightly below that required for a “good enough” screening tool (i.e., AUC = 0.79; Youden’s index = 0.41). Findings from these analyses are presented in Table 6 and Table 7 and Figure 3.

## 4. Discussion

### 4.1. FoB: General Comments

The current study contributes to our general understanding of FoB. First, while similar proportions of nulliparous and multiparous participants met subclinical criteria for a specific phobia, a higher proportion of multiparous participants (5.1%) met full criteria for specific phobia compared to nulliparous participants (1.9%). Thus, a greater proportion of multiparous birthing people reported more distress and impairment related to their FoB symptoms than nulliparous birthing people. Previous research suggests that, overall, nulliparous women may experience higher levels of FoB than multiparous women but that the most severe levels of FoB are experienced by multiparous women [39,47,50,71,72,113]. Furthermore, a history of prior birth experiences, and specifically negative birth experiences, may increase the likelihood of women experiencing more severe FoB in a subsequent pregnancy [39,57,69,70].

Additionally, our study points to important differences between the fear domains most relevant to multiparous and nulliparous birthing individuals. Specifically, for nulliparous participants, fear of cesarean birth and other medical interventions predominated. For multiparous participants, however, a fear of harm to one infant and fear of pain during a vaginal birth emerged. It is likely that the specific fears experienced by multiparous birthing people stem from their previous childbirth experiences. Thus, psychoeducation and interventions given to birthing people suffering from distressing and/or impairing levels of FoB need to take parity into account. Additional research is necessary to further understand how multiparous birthing people’s FoB may be based on realistic fears and experiences (e.g., a knowledge that they are more sensitive to pain or traumatic vaginal birth experiences).

### 4.2. Screening for FoB

In the current study, strong support was found for both the CFQ and the W-DEQ as screening tools for a specific phobia, FoB. Specifically, the CFQ (once specific subscales were identified via logistic regression) and the W-DEQ either met or exceeded the criteria for a “good enough” screening tool across multiple comparisons. These findings provide encouraging support for the CFQ and the W-DEQ as screening tools for diagnosable FoB.

In the first set of analyses of the full measure, the CFQ performed less well than the W-DEQ. However, once the CFQ subscales were selected, using logistic regression, findings strongly supported the use of the CFQ as a screening tool to identify birthing people with subclinical and clinical levels of FoB. Specifically, the findings from individual logistic regression analyses showed the CFQ to perform very well as a screening tool for a specific phobia, FoB. The findings from ROC analyses based on logistic regression showed that the CFQ either fell only slightly short or met or exceeded the criteria for a “good enough” screening tool in all cases. The one exception was for nulliparous participants who were predominantly fearful of vaginal birth. In this case, there were too few positive cases (*n* = 4) for the regression to produce meaningful findings. The full CFQ also met the criteria for a “good enough” screening tool (excluding the positive likelihood ratio) for nulliparous participants primarily fearful of vaginal birth. The CFQ came close to meeting these criteria in three other comparisons: for nulliparous participants in general, for those primarily fearful of cesarean birth, and for those primarily fearful of a vaginal birth (nulliparas and multiparas together).

A number of interesting findings emerged from the logistic regressions of CFQ subscales. Specifically, the CFQ Interference subscale was found to be a robust predictor of specific phobia, FoB across all analyses other than for nulliparous participants primarily fearful of cesarean birth. The CFQ Interference subscale is not part of the Full CFQ, as it specifically assesses impairment and does not measure the intensity of a specific fear domain. It nevertheless appears to be a crucial addition to the measure, allowing for a more sensitive assessment of impairment. The Interference subscale of the CFQ improved the measure’s screening accuracy. This pattern was consistent across evaluations of the CFQ when comparing participants who reported symptoms meeting full diagnostic criteria against all other participants, as well as when comparisons were made with participants reporting symptoms meeting full or subclinical diagnostic criteria against all other participants. This trend remained the case also for analyses examining the full CFQ as well as those employing a subset of the CFQ subscale scores. For any clinical applications of the CFQ as a screening tool for a specific phobia, FoB should include this component of the measure.

Further, the fears of nulliparous participants appear to differ from those of multiparous participants. Specifically, for nulliparous participants, fear of cesarean birth and other medical interventions predominated. For multiparous participants, however, a fear of harm to one infant and fear of pain during a vaginal birth emerged. Among multiparous participants, fear of changes to one’s appearance and sexual functioning, fear of cesarean birth and fear of mom or baby dying were all inversely related to reporting symptoms meeting the criteria for a specific phobia, FoB. Given the multifactorial nature of the CFQ, it appears that specific CFQ subscales or content areas are more relevant to some subgroups of pregnant people based on parity and whether one is more fearful of a vaginal or cesarean birth.

The full CFQ measure performed best when comparing both subclinical and full criteria diagnoses to participants without a diagnosis. The performance of the full CFQ when comparing those who reported symptoms meeting full diagnostic criteria for a specific phobia, FoB, to those who did not report symptoms meeting these criteria was mediocre and felt well below the criteria for a “good enough” screening tool. The screening accuracy of the CFQ was dramatically improved following the use of logistic regression to select a specific CFQ subscale for each subgroup (e.g., nulliparous and multiparous participants). Using specific CFQ subscales to predict diagnostic status resulted in screening metrics that would generally be considered good to excellent. Again, additional research is needed to improve subscales selection for birthing people ONLY meeting full criteria for FoB (as opposed to birthing people experiencing subclinical and clinical symptoms).

Study findings are also consistent with, and build upon, findings from the only other study of the W-DEQ as a screening tool for a specific phobia, FoB [83]. In that previous small (N = 106) study of the screening accuracy of the W-DEQ for a specific phobia, FoB, among nulliparous pregnant people, the W-DEQ evidenced an AUC of 0.96 and a Youden’s index of 0.93. The optimal cut score was determined to be 85. The authors compared participants reporting symptoms meeting full criteria for a specific phobia, FoB, to those who did not. In the present study, the same analysis (i.e., full diagnostic criteria for nulliparous participants only) produced an AUC of 0.88, a Youden’s J index of 0.69, and an optimal cut score of 95.4. Together, these two studies support the screening accuracy of the W-DEQ for a specific phobia, FoB (full criteria). A note of caution regarding these findings is merited given the small numbers of positive cases in both studies, in particular the smaller study by Calderani and colleagues [83].

Our findings suggest that the W-DEQ performs best when comparing pregnant people who have reported symptoms meeting full diagnostic criteria for FoB to those who did not report symptoms meeting these criteria. A note of caution here is also merited due to the fact that the number of participants meeting the full criteria was small, rendering estimates of performance unstable. Additional research involving larger samples is needed to fully clarify the merits and disadvantages of screening for a specific phobia, FoB full criteria versus full or subclinical, and to ensure the stability and replicability of estimates of performance, especially for comparisons of specific phobia, FoB full diagnostic criteria to all other participants.

Interestingly, when we compared participants who reported symptoms meeting full or subclinical diagnostic criteria for a specific phobia, FoB, to the remainder of the sample, the W-DEQ performed best when limiting these analyses to participants who were primarily fearful of vaginal birth. It may be that the W-DEQ performs best for people who are most fearful of vaginal birth, but additional research will be needed to clarify this. Of note, when limiting the analysis to those primarily fearful of vaginal birth, the W-DEQ performed best for multiparous participants. This is counter-intuitive in that one might expect the fears of multiparous people to more closely resemble symptoms of post-traumatic stress disorder and not specific phobia [49,67].

### 4.3. Limitations and Future Directions

Although this study was adequately powered (N = 659), subsamples of participants reporting symptoms meeting full diagnostic criteria for a specific phobia, FoB, were much smaller. Consequently, we were unable to conduct all ROC analyses comparing participants whose symptoms met the full criteria for a specific phobia, FoB, against the remaining participants. For some ROC analyses, we compared those who reported symptoms meeting full or subclinical criteria against the remaining participants. This improved power but may not fully generalize to pregnant people with symptoms meeting full criteria for specific phobia FoB. Future research with larger samples will be able to refine some of the findings from the present research.

Given that specific phobia may not be the only diagnostic category most relevant for FoB, it would be extremely helpful to evaluate the ability of the CFQ and the W-DEQ to screen for any mental health diagnosis under which a particular person’s FoB may fall. For example, for some people, FoB may be best characterized as a post-traumatic stress disorder, whereas for others, it may be best understood as a specific phobia or health anxiety. It would be helpful to know if the majority of people whose FoB is severe enough to merit a mental health diagnosis can be captured by the CFQ or the W-DEQ. Studies in which the screening ability of these two measures are assessed against a broader range of anxiety-related conditions will be able to answer this question.

Future research may benefit from efforts to replicate the regression analyses and resulting ROC findings of the CFQ subscales to ensure the stability of these findings. Future research will also be needed to ascertain the utility of the CFQ and W-DEQ in diverse cultural groups, social contexts (e.g., lower socio-economic status) and countries.

## 5. Conclusions

The W-DEQ performs well as a screening tool for a specific phobia, FoB, for pregnant people overall and across various subgroups (e.g., nulliparous and multiparous pregnant people). The CFQ performs less well as a screening tool for a specific phobia, FoB, but nevertheless holds promise. Additional research is needed to ensure replicability of findings and to further evaluate the potential of the CFQ to accurately screen for diagnosable FoB.

## Figures and Tables

**Figure 1 ijerph-19-04647-f001:**
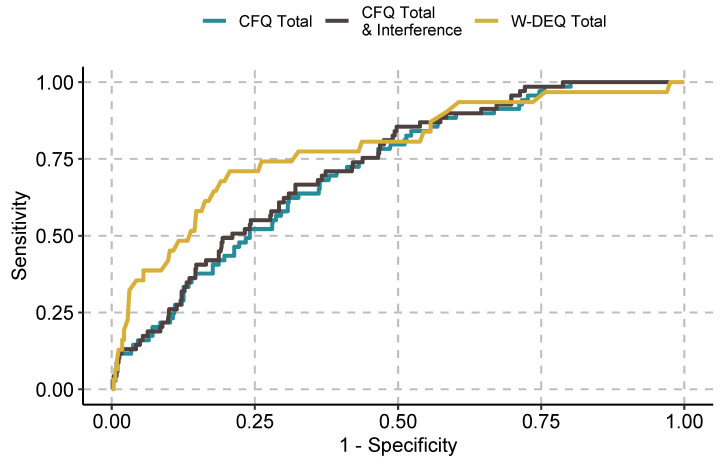
Receiver operating characteristic (ROC) curves for the Childbirth Fear Questionnaire (CFQ) and the Wijma Delivery Expectations Questionnaire (W-DEQ) for the full sample (full diagnostic criteria ONLY).

**Figure 2 ijerph-19-04647-f002:**
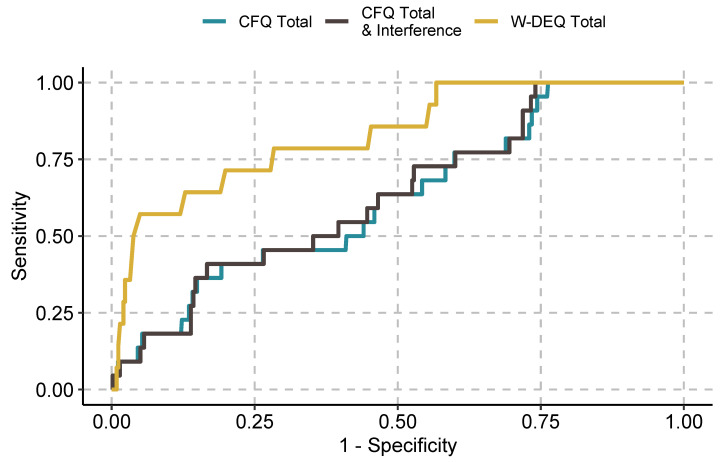
ROC curves for the CFQ and the W-DEQ for the full sample (subclinical and full diagnostic criteria combined).

**Figure 3 ijerph-19-04647-f003:**
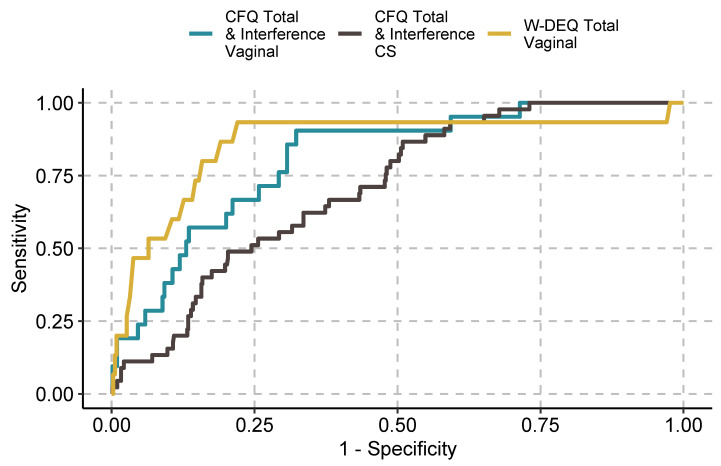
ROC curves for the CFQ (Total and Interference subscale scores) and the W-DEQ (subclinical and full diagnostic criteria combined).

**Table 1 ijerph-19-04647-t001:** Participant demographic information and reproductive information (N = 659).

Demographic Variables		
	Percentage	n
Married or cohabitating	93.3%	613
Cis-gender female	99.1%	652
Some postsecondary education	94.4%	623
European heritage	76.3%	502
English spoken at home	95.4%	629
**Current Pregnancy**		
Singleton pregnancy	97.7%	642
Weeks pregnant: M (SD)	34.6 (2.1)	497
Pregnancy complications	30.8%	202
**Reproductive History**		
Prior births	52.3%	296
Prior vaginal birth	51.2%	198
Prior cesarean birth	17.7%	66
Prior pregnancy loss < 20 weeks	40.6%	157
Prior pregnancy loss > 20 weeks	1.3%	5

**Table 2 ijerph-19-04647-t002:** Means (M) and standard deviations (SD) for the Childrbith Fear Questionnaire (CFQ; total and subscales) and the Wijma Delivery Expectations Questionnaire (W-DEQ).

	Full Sample M (SD)	Nullips Only M (SD)	Multips Only M (SD)
CFQ Total	1.11 (0.59)	1.23 (0.61)	1.02 (0.56)
CFQ Interference	0.42 (0.47)	0.44 (0.47)	0.39 (0.45)
W-DEQ	55.44 (23.76)	59.07 (22.77)	52.8 (24.09)

Note: CFQ Total and CFQ Interference scores are mean items scores (i.e., out of a possible 0–4). W-DEQ scores are for the total out of 33 items.

**Table 3 ijerph-19-04647-t003:** Receiver operating characteristic (ROC) results for the Childbrith Fear Questionnaire (CFQ) and the Wijma Delivery Expectations Questionnaire (W-DEQ) across parity (full diagnostic criteria ONLY).

		Prevalence	AUC	*J*	Cutpoint	Sensitivity	Specificity	NPV	LR+
CFQ Total Scores	Full sample	3.3%	0.63	0.11	1.17	0.56	0.55	0.97	1.24
	Nulliparous only	1.9%	0.45	0.45	1.34	0.60	0.45	0.98	1.09
	Multiparous only	5.0%	0.67	0.17	1.05	0.60	0.57	0.96	1.40
CFQ Total & Interference Subscale Scores	Full sample	3.3%	0.62	0.10	1.13	0.53	0.57	0.97	1.23
	Nulliparous only	1.9%	0.56	0.23	0.69	1.00	0.23	1.0	1.30
	Multiparous only	5.0%	0.69	0.35	1.63	0.47	0.89	0.97	4.27
W-DEQ	Full sample	3.9%	0.82	0.43	78.87	0.62	0.81	0.98	3.26
	** *Nulliparous only* **	** *2.5%* **	** *0.88* **	** *0.69* **	** *95.37* **	** *0.75* **	** *0.94* **	** *0.99* **	** *12.50* **
	** *Multiparous only* **	** *5.9%* **	** *0.83* **	** *0.53* **	** *76.56* **	** *0.70* **	** *0.83* **	** *0.98* **	** *4.12* **

Note: Cut scores for the CFQ are mean items scores (i.e., out of a possible 0–4). W-DEQ cut scores are for the total out of 33 items; AUC = Area under the curve; *J* = Youden’s J Index; NPV = Negative predictive value; LR+ = Positive likelihood ratio.

**Table 4 ijerph-19-04647-t004:** ROC Results for the Childbrith Fear Questionnaire (CFQ) and the Wijma Delivery Expectations Questionnaire (W-DEQ) across parity (subclinical and full diagnostic criteria combined).

		Prevalence	AUC	*J*	Cutpoint	Sensitivity	Specificity	NPV	LR+
CFQ Total Scores	Full sample	10.0%	0.72	0.29	1.18	0.29	0.69	0.90	0.94
	Nulliparous only	8.0%	0.75	0.30	1.46	0.66	0.64	0.96	1.83
	Multiparous only	12.0%	0.71	0.26	1.13	0.63	0.63	0.93	1.70
CFQ Total & Interference Subscale Scores	Full sample	10.0%	0.73	0.30	1.13	0.69	0.61	0.95	1.77
	Nulliparous only	8.0%	0.77	0.37	1.38	0.71	0.66	0.96	2.09
	Multiparous only	12.0%	0.73	0.30	1.05	0.67	0.63	0.93	1.81
W-DEQ Total Scores	Full sample	9.0%	0.79	0.47	73.59	0.68	0.79	0.96	3.24
	Nulliparous only	7.0%	0.68	0.26	81.51	0.44	0.82	0.95	2.44
	** *Multiparous only* **	** *11.0%* **	** *0.88* **	** *0.53* **	** *70.62* **	** *0.74* **	** *0.79* **	** *0.96* **	** *3.52* **

Note: Cut scores for the CFQ are mean items scores (i.e., out of a possible 0–4). W-DEQ cut scores are for the total out of 33 items; AUC = Area under the curve; *J* = Youden’s J Index; NPV = Negative predictive value; LR+ = Positive likelihood ratio.

**Table 5 ijerph-19-04647-t005:** ROC Results for the Childbrith Fear Questionnaire (CFQ; Total and Interference subscale scores) and the Wijma Delivery Expectations Questionnaire (W-DEQ), separately for fear of vaginal and fear of cesarean birth (subclinical and full diagnostic criteria combined).

**CFQ Total & Interference Subscale Scores**									
		**Prevalence**	**AUC**	** *J* **	**Cutpoint**	**Sensitivity**	**Specificity**	**NPV**	**LR+**
Fear of Vaginal Birth	Full sample	3.2%	0.81	0.43	1.38	0.71	0.72	0.99	2.54
	Nulliparous only	** *1.5%* **	** *0.88* **	** *0.67* **	** *1.42* **	** *1.00* **	** *0.67* **	** *1.00* **	** *3.03* **
	Multiparous only	4.1%	0.80	0.44	1.38	0.67	0.77	0.98	2.91
Fear of cesarean birth	Full sample	6.9%	0.71	0.27	1.04	0.73	0.54	0.96	1.59
	Nulliparous only	4.9%	0.78	0.49	1.51	0.77	0.72	0.98	2.75
	Multiparous only	8.6%	0.73	0.39	0.94	0.84	0.55	0.97	1.87
**W-DEQ**									
		**Prevalence**	**AUC**	** *J* **	**Cutpoint**	**Sensitivity**	**Specificity**	**NPV**	**LR+**
Fear of Vaginal Birth	Full sample	** *4.2%* **	** *0.86* **	** *0.56* **	** *78.87* **	** *0.74* **	** *0.83* **	** *0.99* **	** *4.35* **
	Nulliparous only	2.5%	0.73	0.70	96.36	0.75	0.95	0.99	15.0
	Multiparous only	** *5.3%* **	** *0.92* **	** *0.70* **	** *75.24* **	** *0.89* **	** *0.81* **	** *0.99* **	** *4.68* **

AUC = Area under the curve; *J* = Youden’s J Index; NPV = Negative predictive value; LR+ = Positive likelihood ratio.

**Table 6 ijerph-19-04647-t006:** Results of logistic regressions on Chidbirth Fear Questionnaire (CFQ) subscales for nulliparous participants.

	SP Diagnostic Status Dichotomized (FULL&SUB versus NOT)–Reduced Model			Fear of CS Birth Dichotomized (FULL&SUB versus NOT)–Reduced Model		
*Predictors*	*Log-Odds*	*CI*	*p*	*Log-Odds*	*CI*	*p*
(Intercept)	−6.23	−8.18–−4.69	<0.001	−8.96	−13.09–−6.18	<0.001
INT	1.31	0.48–2.22	0.003	1.31	0.32–2.43	0.01
CS	0.62	0.09–1.19	0.03	1.57	0.68–2.71	0.002
INTERFERENCE	1.01	0.08–1.97	0.03			
Observations	267			267		
R^2^ Tjur	0.25			0.27		
AUC	0.87	Cases correctly classified: 16/22 positive cases 206/245 negative cases		0.94	Cases correctly classified: 11/13 positive cases 214/254 negative cases	
Optimal cutpoint	0.10			0.12		
Youden’s index	0.51			0.65		
Sensitivity	0.69			0.80		
Specificity	0.82			0.85		

Note: Formulas for predicted probability for individual (i): $P\left({\mathrm{FoB}}_{i}\right)=\frac{1}{1+e^{-}}$, $P\left({\mathrm{FoCB}}_{i}\right)=\frac{1}{1+e^{-}}$; Full = Full clinical diagnostic criteria; SUB = Subclinical diagnostic criteria; SP = Specific phobia; CI = Confidence interval; AUC = Area under the curve; FoB = Fear of childbrith; FoCB = Fear of cesarean birth; INT = Fear of medical intervention; CS = Fear of ceserean section.

**Table 7 ijerph-19-04647-t007:** Results of logistic regressions on CFQ subscales for multiparous participants.

	SP Diagnostic Status Dichotomized (FULL&SUB versus NOT)–Reduced Model			Fear of Vaginal Birth Dichotomized (FULL&SUB versus NOT)–Reduced Model			Fear of Cesarean Birth Dichotomized (FULL&SUB versus NOT)–Reduced Model		
*Predictors*	*Log-Odds*	*CI*	*p*	*Log-Odds*	*CI*	*p*	*Log-Odds*	*CI*	*p*
(Intercept)	−3.83	−4.86–−2.96	<0.001	−5.89	−8.38–−4.02	<0.001	−3.25	−4.07–−2.54	<0.001
SEX	−1.1	−1.96–−0.36	0.007				−0.72	−1.59–0.01	0.074
PAIN	0.76	0.33–1.21	0.001	1.03	0.39–1.75	0.003			
HARM				1.29	0.08–2.61	0.044			
CS				−0.79	−1.65–−0.05	0.049			
DEATH				−1.02	−2.18–−0.01	0.063			
INTERFERENCE	2.48	1.66–3.39	<0.001	2.43	1.09–4.00	0.001	2.31	1.48–3.21	<0.001
Observations	291			291			291		
R^2^ Tjur	0.24			0.239			0.16		
AUC	0.84	Cases correctly classified: 21/34 positive cases 215/257 negative cases		0.92	Cases correctly classified: 9/12 positive cases 249/279 negative cases		0.79	Cases correctly classified: 17/25 positive cases 214/266 negative cases	
Optimal cutpoint	0.15			0.07			0.10		
Youden’s index	0.42			0.67			0.41		
Sensitivity	0.62			0.77			0.61		
Specificity	0.82			0.90			0.80		

Formula for predicted probability for individual (i): $P\left({\mathrm{FoB}}_{i}\right)=\frac{1}{1+e^{-\left(-3.83-1.1\cdot SEX_{i}+0.76\cdot PAIN_{i}+2.48\cdot Interference_{i}\right)}}$, $P\left({\mathrm{FoVB}}_{i}\right)=\frac{1}{1+e^{-\left(-5.89+1.03\cdot PAIN_{i}+1.29\cdot HARM_{i}-0.79\cdot CS_{i}-1.02\cdot DEATH_{i}+2.43\cdot Interference_{i}\right)}}$, $P\left({\mathrm{FoCB}}_{i}\right)=\frac{1}{1+e^{-\left(-3.25-0.72\cdot SEX+2.31\cdot Interference_{i}\right)}}$; Full = Full clinical diagnostic criteria; SUB = Subclinical diagnostic criteria; SP = Specific phobia; CI = Confidence interval; AUC = Area under the curve; FoB = Fear of childbrith; FoVB = Fear of vaginal birth; FoCB = Fear of ceserean birth; INT = Fear of medical intervention; SEX = fear of loss of sexual pleasure or attractiveness; PAIN = fear of pain from a vaginal birth; HARM = fear of harm to the baby; CS = Fear of cesarean section; DEATH = fear of mom or baby dying.

## Data Availability

The datasets used and/or analyzed during the current study are available from the corresponding author on reasonable request.

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
