# Peer review of "The Childbirth Fear Questionnaire and the Wijma Delivery Expectancy Questionnaire as Screening Tools for Specific Phobia, Fear of Childbirth"

_ijerph, 2022, doi:10.3390/ijerph19084647_

Round 1
Reviewer 1 Report
Review MDPI IJERPH 2022.02.25
Thank you so much for the opportunity to review this manuscript. Overall, the CFQ the authors developed seems to be very useful in covering the limitation of WDEQ assessing both fear of vaginal childbirth as well as fear of cesarean section.
Overall, the manuscript was well-written with literature review in the context of psychiatric diagnosis, however the introduction part is too long to understand the points. Unfortunately, there are some of the points unclear regardless of the long introduction.
Here are my detailed comments.
Major parts:
- More justification is needed why the authors insisted that fear of childbirth is consistent with specific phobia in DSM-5, and why DARTS (DSM5) will be the gold standard for assessing fear of childbirth. According to the DSM5, specific phobia is a fear towards circumscribed fear. So, is childbirth as ‘circumscribed subject”? (L100) Also, the diagnostic category of FOB is still controversial; some say specific phobia whereas others might say PTSD (experience of miscarriage, or abuse).
- ROC:
Why the authors did not exclude some of the items of CFQ in order to obtain a better score of ROC. Statistical re-consideration will be required.
Detailed parts:
Introduction
- Maternal prenatal anxiety disorder is more common than depressive symptoms. However, it is unlikely to agree if there is strong evidence the anxiety condition is more associated with preeclampsia, low birth weight, adverse effects on an infant's brain, from the following sentence. The association may depend on the severity of continuing period of an anxiety condition, influenced by other confounding factors. Hence, the sentence sounds too strong. I would recommend authors to revise the following part (less judgmentally).
Maternal prenatal anxiety (i.e., dimensional anxiety not necessarily associated with a 54 diagnosis) is associated with numerous adverse pregnancy outcomes such as preterm de-55 livery, miscarriage, preeclampsia, and low birth weight [14–18], as well as prolonged neg-56 active effects on the developing infant (e.g., impaired brain activity, difficult temperament, 57 impaired self-regulation and motor development, and an increased risk for attention-def-58 icit/hyperactivity disorder) [15,19–24].
- Line 70-72: Please insert references in each mental disorder.
- Overall, the introduction part is high volume. Hence, I would recommend authors shorten the paragraph (Line 115-149) integrating it to the former paragraph (Line 75-92). Paragraphs (150-160) can also be shortened and summarized.
- This part can be moved from intro to methods for a simple introduction.
They propose that, in order 189 for a screening tool to be deemed sufficiently accurate for use in clinical settings, it should 190 meet certain minimum standards of accuracy including an area under the curve (AUC) of 191 .8 or greater, a Youden’s J index of .5 or more (J = .05 when sensitivity and specificity both 192 equal .75), a negative predictive value (NPV) of .8 or greater, and a positive likelihood 193 ratios (LR+) of 4.0 or more. An LR+ of 4.0 means that with a positive test result, one is 25% 194 more likely to have the condition in question, compared with the baseline probability of 195 having the condition [98].
Methods;
- The following terms are unclear in the methods. Clarification/more explanation is needed.
- ‘Subclinical and full criteria diagnoses of specific phobia, FoB’.
- ‘Interference subscales’(Exactly what kind of items were made, the number of items)
- good enough scale
- CFQ& interference subscales (I guess the authors add total items of CFQ with items of interference
subscales. )
- In order to get cut-off-point (ROC, AUC). I guess The Diagnostic Assessment Research Tool was used as a gold standard measure to identify the prevalence, sensitivity, and specificity of each WDEQ & CFQ. Please add more explanation in the analysis
- Please mention the range of scores of total CFQ items (47) or subscales. Why the cut-off score was very small(1.1X in Table 4?)?
- Justification of adopting subclinical diagnosis criteria of FOC will be needed.
Results
- My suggestion is to omit some of the figures or move some to an appendix.
Discussion.
- Justification of adopting subclinical diagnosis criteria of FOC will be needed.
- I assume that ROC is low because the CFQ includes various concepts such as fear of caesarean section, etc this may be one of the reasons for inconsistency with the DSM-5. I believe that more discussion for the lower ROC of CFQ based on comparison of subscales of CFQ with DSM-5 diagnostic tool will be required.
Hope my comments will be helpful to the authors’ revision.
Reviewer 2 Report
See Attach files
